# Application of machine learning and complex network measures to an EEG dataset from ayahuasca experiments

**Caroline L. Alves**[1,2]*, **Rubens Gisbert Cury**[3], **Kirstin Roster**[2], **Aruane M. Pineda**[2], **Francisco A. Rodrigues**[2], **Christiane Thielemann**[1], **Manuel Ciba**[1]

**1** BioMEMS Lab, Aschaffenburg University of Applied Sciences (UAS), Aschaffenburg, Germany, **2** Institute of Mathematical and Computer Sciences, University of São Paulo (USP), São Paulo, Brazil, **3** Department of Neurology, Movement Disorders Center, University of São Paulo (USP), São Paulo, Brazil

\* Caroline.Alves@th-ab.de

**Data Availability Statement:** The data used in this work may be obtained here: Ekman Schenberg, Eduardo, 2015, "Acute Biphasic effects of

## Abstract

Ayahuasca is a blend of Amazonian plants that has been used for traditional medicine by the inhabitants of this region for hundreds of years. Furthermore, this plant has been demonstrated to be a viable therapy for a variety of neurological and mental diseases. EEG experiments have found specific brain regions that changed significantly due to ayahuasca. Here, we used an EEG dataset to investigate the ability to automatically detect changes in brain activity using machine learning and complex networks. Machine learning was applied at three different levels of data abstraction: (A) the raw EEG time series, (B) the correlation of the EEG time series, and (C) the complex network measures calculated from (B). Further, at the abstraction level of (C), we developed new measures of complex networks relating to community detection. As a result, the machine learning method was able to automatically detect changes in brain activity, with case (B) showing the highest accuracy (92%), followed by (A) (88%) and (C) (83%), indicating that connectivity changes between brain regions are more important for the detection of ayahuasca. The most activated areas were the frontal and temporal lobe, which is consistent with the literature. F3 and PO4 were the most important brain connections, a significant new discovery for psychedelic literature. This connection may point to a cognitive process akin to face recognition in individuals during ayahuasca-mediated visual hallucinations. Furthermore, closeness centrality and assortativity were the most important complex network measures. These two measures are also associated with diseases such as Alzheimer's disease, indicating a possible therapeutic mechanism. Moreover, the new measures were crucial to the predictive model and suggested larger brain communities associated with the use of ayahuasca. This suggests that the dissemination of information in functional brain networks is slower when this drug is present. Overall, our methodology was able to automatically detect changes in brain activity during ayahuasca consumption and interpret how these psychedelics alter brain networks, as well as provide insights into their mechanisms of action.

ayahuasca", https://doi.org/10.7910/DVN/VVE6QC, Harvard Dataverse, V1 This information may be found here: Schenberg EE, Alexandre JFM, Filev R, Cravo AM, Sato JR, Muthukumaraswamy SD, et al. (2015) Acute Biphasic Effects of Ayahuasca. PLoS ONE 10(9): e0137202. https://doi.org/10.1371/journal.pone.0137202 doi: 10.1371/journal.pone.0137202 We also uploaded the Pearson connectivity matrix from the original data in the following Figshare repositories here: Pearson's connection matrix of the EEG experiments of subjects who ingested Ayahuasca at the time after the psychedelic activation time can be found here: Alves, Caroline (2022): With-ayahuasca. Figshare repository. Dataset. https://doi.org/10.6084/m9.figshare.21082513.v1 Pearson's connection matrix of the EEG experiments of subjects who ingested Ayahuasca at the time before the psychedelic activation time: Alves, Caroline (2022): No-ayahuasca. Figshare repository. Dataset. https://doi.org/10.6084/m9.figshare.21082531.v1.

**Funding:** This study was financially supported by Conselho Nacional de Desenvolvimento Científico e Tecnológico (CNPq) in the form of a grant awarded to FAR (309266/2019-0). This study was also financially supported by Fundação de Amparo à Pesquisa do Estado de São Paulo (FAPESP) in the form of grants awarded to FAR (19/23293-0), AMP (2019/22277-0) and KR (2019/26595-7). The funders had no role in study design, data collection and analysis, decision to publish, or manuscript preparation.

**Competing interests:** The authors have declared that no competing interests exist.

# 1 Introduction

Ayahuasca is made from a blend of Amazonian herbs [1]. This combination of plants is often associated with rituals of different religions and social groups. Ayahuasca has been used in the Amazon for a couple of hundred years, being part of the traditional medicine of the indigenous population within this region [2].

Since the use of ayahuasca has spread throughout many countries, it is necessary to study in depth its cerebral mechanisms and its potential clinical implications. In addition, because it affects brain areas related to emotions, memories, and executive functions, ayahuasca might be used in the treatment of psychiatric disorders, such as drug addiction [3–5], Parkinson's disease [6–9], and depression [10–16]. For example, an open-label clinical study found significant therapeutic benefits among patients with treatment-resistant major depressive disorder after the administration of a single dose of ayahuasca [12]. Moreover, a randomized trial showed that ayahuasca doses were associated with reductions in depressive symptoms in patients with major depressive disorder, compared to placebo treatments [11].

Additionally, ayahuasca has been shown to elicit anti-neuroinflammatory properties [16] and stimulate adult neurogenesis in vitro [17]. In this line, ayahuasca could be helpful for the treatment of several neurological diseases well known to harbor inflammation in its physiopathology [18], including chronic degenerative diseases and illnesses related to acute injury, such as cerebral ischemia, multiple sclerosis, and Alzheimer's disease (AD) [19, 20].

The EEG data studied here are from [21], from subjects who ingested ayahuasca. This study observed slow-gamma power increases at the left Centro-parietal-occipital, left frontotemporal, and right frontal cortices. In contrast, fast-gamma increases were significant at the left Centro-parieto-occipital, left frontotemporal, right frontal, and right parieto-occipital cortices due to ayahuasca ingestion. As a result, this study concentrated solely on the changes in frequency bands caused by the use of the psychedelic substance.

Despite the enormous therapeutic potential of ayahuasca, in most countries, it is an illegal substance and only legalized for religious use, such as in Brazil. Therefore, few studies on human beings are found in the literature, and more research is needed on how this substance alters the brain and its mechanism of action.

The use of graph theory mathematical approaches gave intriguing insights into the intricate network structure of the human brain, which is also related with pathological states [22–25]. Notably, complex networks have been employed as biomarkers for a variety of disorders [26, 27]. Furthermore, the community detection algorithm (also referred to as the clustering graph) is a fundamental analysis technique that aims to identify densely connected structures within complex networks [28–30]. Several studies have used complex network measurements and community detection algorithms to detect brain activity in EEG data recently [31–33]

Because of the increased amount of data related to health, such as medical records, exams of patients, and hospital resources, machine learning (ML) algorithms have become more applicable, primarily for medical diagnosis [34–37], in order to provide more accurate and automatic investigations of various diseases [38] and may be an important tool capable of detecting acute and permanent abnormalities in the brain. In addition, many studies have utilized machine learning algorithms to capture brain activity using raw EEG time series [39, 40], the correlation between electrodes [41, 42], and complex network measures [23].

Also, in contrast to traditional statistical methods, the ML approach has the advantage that it does not rely on prior assumptions (such as adequate distribution, independence of observations, absence of multicollinearity, and interaction problems) and is also well suited to analyze and capture complex nonlinear relationships in data automatically. Nevertheless, new techniques have emerged to assist in interpreting machine learning results, e.g., SHapley Additive

Explanations (SHAP) values. Any machine learning algorithm may use this metric for identifying and prioritizing features [43–45].

The purpose of this study is to determine whether it is possible to automatically detect the changes in brain activity after intake of ayahuasca with machine learning methods using the following data abstraction levels for the input: (A) raw EEG time series, (B) the correlation between the EEG electrodes as used in (A) represented by a connectivity matrix, and (C) complex network measures extracted from (B). In contrast to articles in the literature that use only one of these levels of abstraction, this study uses all three levels. In addition, we define which of these abstraction levels is most appropriate for capturing ayahuasca-induced brain changes. The SHAP value method has also been shown to be more effective than the studies cited above in identifying the best brain regions, the best connections between the brain regions, and the best measures of complex networks, which can be used to interpret the effects of the psychedelic substance on the brain. A final result of this research was the creation of new measures that have never been used before within the literature, which can be used as input to machine learning algorithms to assess the size of community structures.

## 2 Materials and methods

The python code used for the analysis is available at https://github.com/Carol180619/Paper-ayahuasca.git.

### 2.1 Data

The data used for this study has been made openly available by the Federal University of São Carlos, Brazil [21]. Sixteen healthy male and female patients with prior ayahuasca experience (eight women, mean 29.0 years; 12 men, mean 38.5 years) agreed (with written permission) to consume this psychedelic substance while EEG recordings were made (The following exclusion criteria were used: minors than the age of 21 years, personal history of psychiatric illness, current use of any psychiatric medication, cardiovascular disease, and any neurological disorders or brain damage in the previous year). All methodologies for this investigation were approved by the Universidade Federal de São Paulo's Ethical Committee, and the study was carried out in compliance with available criteria for human hallucinogen research safety [46].

Patients were instructed to close their eyes and remain in a resting condition. A nurse accompanied the experiment for its duration of 225 minutes. The recordings began 25 minutes before ayahuasca consumption and ended 200 minutes afterward. The main compounds in the brew were [21]: Dimethyltryptamine (DMT), DMTN-oxide (DMT-NO), N-methyltryptamine (NMT), indoleacetic acid (IAA), 5-hydroxy-DMT (5-OH-DMT, or bufotenin), 5-methoxy-DMT (5-MeO-DMT), Harmine, Harmol, Harmaline, Harmalol, THH, 7-hydroxy-tetrahydroharmine (THH-OH), and 2-methyl-tetrahydro-beta- carboline (2-MTHBC). All recordings were downsampled to 500 Hz, bandpass filtered between 0.5 and 150 Hz, and artifacts due to movements were removed. Recordings were made with 62 electrodes, following the EEG electrode positions in the 10–10 system. These channels are: Fp1, Fz, F3, F7, FT9, FC5, FC1, C3, TP9, CP5, CP1, Pz, P3, P7, O1, Oz, P8, TP10, CP6, CP2, C4, T8, FT10, FC6, FC2, F4, F8, Fp2, AF7, AF3, AFz, F1, F5, FT7, FC3, FCz, C1, C5, TP7, CP3, P1, P5, PO7, PO3, POz, PO4, PO8, P6, P2, CPz, CP4, TP8, FC4, FT8, F6, F2, AF4, AF8, O2, P4, C6, and C2 (see in Appendix A (Fig 13) of S1 Appendix). It is worth mentioning that after using ayahuasca, all individuals experienced notable alterations in their typical state of consciousness.

Further details are given in [21].

## 2.2 Machine learning algorithm

**2.2.1 Classification.** In order to classify the (A) EEG time series, (B) the connectivity matrices, and (C) the complex network measures, the support vector machine (SVM) [47] algorithm was used. SVM has been used with superior results for the classification of complex network measures before by other groups [48–50] and performed superior in our comparative evaluation. In this analysis, we compared the following machine learning methods to classify the complex network measures: Random forest (RF) [51], SVM [47], naive bayes (NB) [52], multilayer perceptron (MLP) [53], stochastic gradient descent with linear models classifier (SGD) [54], logistic regression (LR) [55] and extreme Gradient Boosting classifier [56] (XGBoost). The results can be found in Appendix C in S1 Appendix.

A more robust deep learning (DL) algorithm from [41] (in which the model was named tuned convolutional neural network) was also tested. The results using DL are in the Appendix D in S1 Appendix.

**2.2.2 Resampling and evaluation.** The dataset was resampled by separating it into training (train) and test sets, with 25% of data composing the test set. Then, for a reliable model, a k-cross validation was used [57], with k = 10 (value widely used in the literature [58–62]). A hyper-parameter optimization called grid search was used here, similar to [63–67]. The hyper-parameter optimization values used for each classifier models can be found in Appendix C in S1 Appendix.

For evaluation, accuracy (Acc.) was used as the standard performance metrics, as is the state-of-art in the literature [37, 68–71]. Since the problem here is a two-class (negative and positive) classification problem, other metrics considered here are the measures of precision and recall, also commonly used in the literature [72–75]. Precision (also called positive predictive value) is the proportion of relevant instances among those retrieved. Whereas recall (also called sensitivity) measures how well a classifier can predict positive examples (hit rate in the positive class), here related with an effect of the ayahuasca. Another measure used here and also used in literature [64, 76, 77] is the F1 score which is the harmonic mean of the recall and precision [78]. For visualization of these two latter measures, the receiver operating characteristic (ROC) curve is a standard method as it displays the relation between the rate of true positives and false positives. The area below this curve, called the area under the ROC curve (AUC), has been widely used in classification problems [66, 68, 79, 80]. The value of the AUC varies from 0 to 1, where the value of one corresponds to a classification result free of errors. $AUC = 0.5$ indicates that the classifier is not able to distinguish the two classes; this result is equal to the random choice. Furthermore, we consider the micro average of the ROC curve, which computes the AUC metric independently for each class (calculate AUC metric for healthy individuals, class zero, and separately calculate for unhealthy subjects, class one), and then the average is computed considering these classes equally. The macro average is also used in our evaluation, which does not consider both classes equally, but aggregates the contributions of the classes separately and then calculates the average.

Furthermore, we interpret the machine learning results using SHapley Additive exPlanations (SHAP) values [81] to quantify the importance of the complex measures, connections of brain regions, and location of electrodes for the classification result. This metric enables the identification and prioritization of features and can be used with any machine learning algorithm [43–45].

## 2.3 Input data for machine learning

The following three data abstraction levels were applied to a classification algorithm as described in subsection 2.2 Machine learning algorithm: (2.3.1 EEG time series) the EEG time

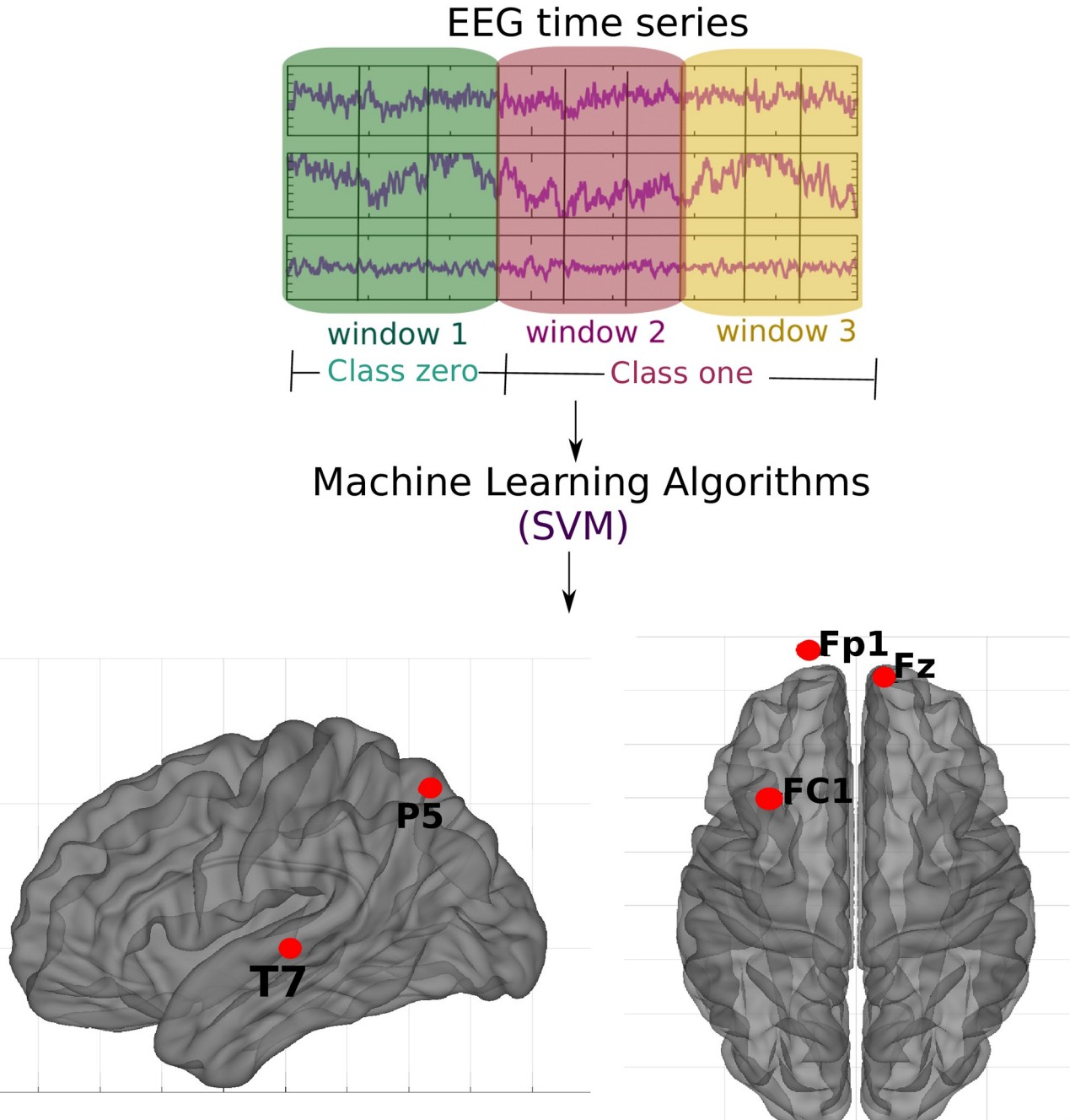

**Fig 1. Methodology of the subsection using raw EEG time series.** For each participant, the EEG time series was split into three parts. Those corresponding to the first window were labeled as class 0 (no effect of ayahuasca) and those corresponding to the second and third windows as class 1 (under the influence of ayahuasca), and then SVM was used. The objective was to determine which brain parts are most influenced by ayahuasca consumption. The crucial areas discovered using the SHAP values approach are emphasized in the illustration.

series (Fig 1), (2.3.2 Connectivity matrices) the connectivity matrix calculated by means of the Pearson correlation of the EEG time series (Fig 2), and (2.3.3 Complex network measures) the complex network measures calculated from the connectivity matrix (Fig 3).

**2.3.1 EEG time series.** The data was divided into three "time windows" (see Table 1). The first window (25 minutes before ingestion until 50 minutes after ingestion of ayahuasca) was

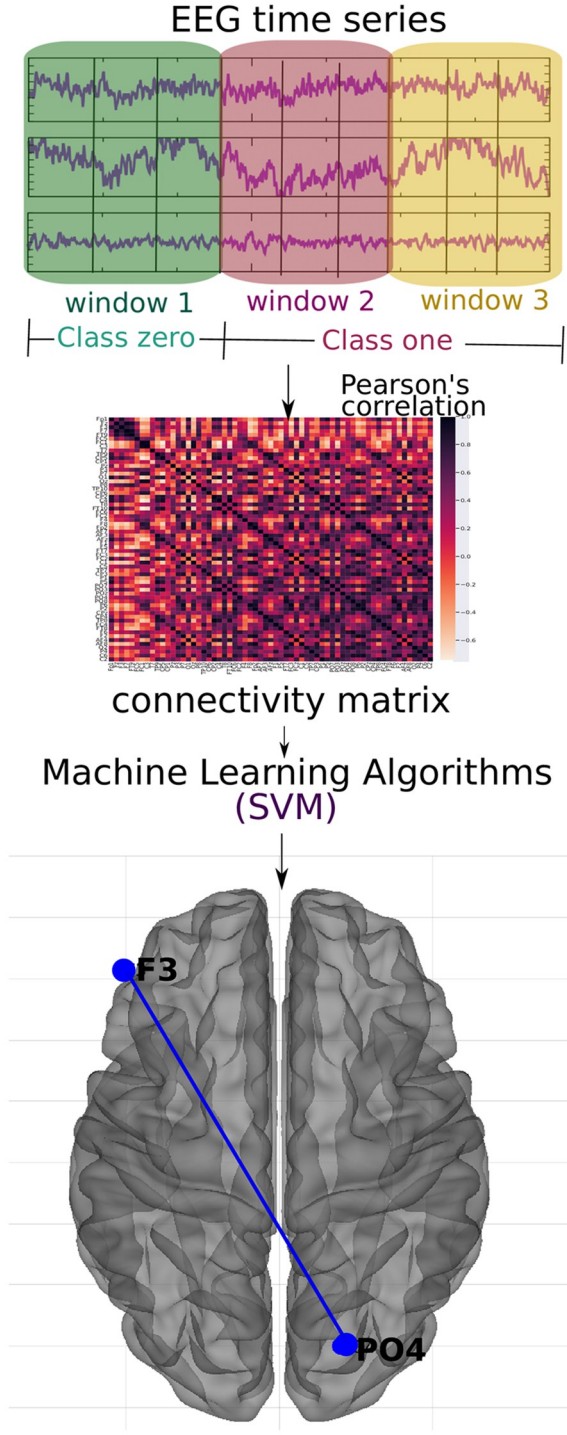

**Fig 2. Methodology of the subsection using connectivity matrices.** For each of the time windows, the Pearson correlation connectivity matrix was generated, and then they were classified with the SVM method considering the first window as zero label (without ayahuasca) and the other two as one label (with ayahuasca). This analysis aimed to verify the best connections of the brain areas used during ayahuasca use. The principal connection discovered using the SHAP value approach is depicted in the picture.

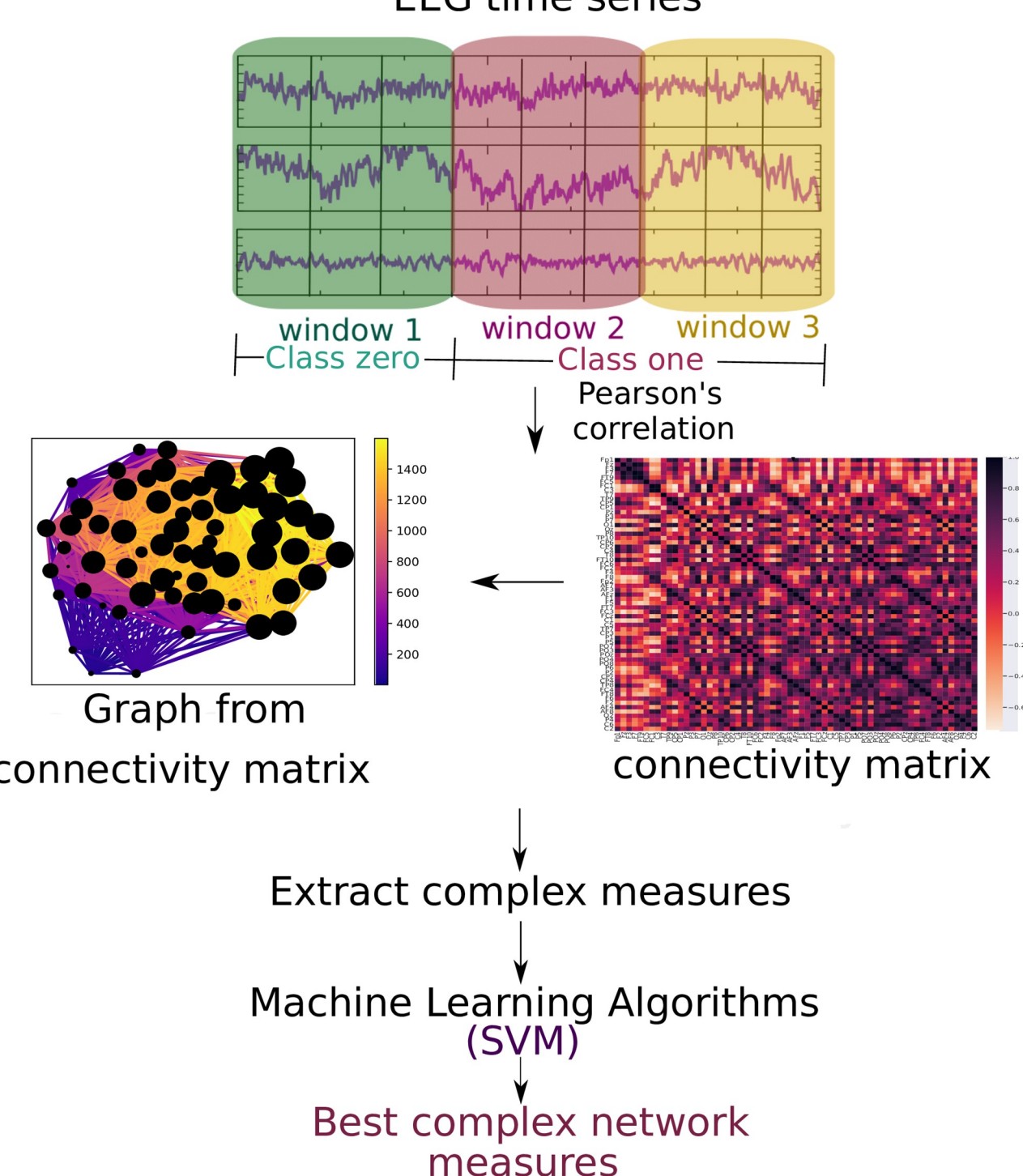

**Fig 3. Methodology of the subsection using complex network measures.** The EEG time series is divided into three parts. For each of them, the Pearson correlation was calculated. For each window, a connectivity matrix was generated (in the Fig, the connectivity matrix of the first window of the first subject containing the 62 electrodes, the color bar containing the connection strength between these electrodes). A graph was formed for each of them (in the Fig, the graph of this connectivity matrix has 62 nodes and the connection strength according to the color bar and the node size according to its number of connections), and complex network measures are extracted from them.

**Table 1. Definition of time windows of the EEG signal.** Window 1 is considered the control (without effect of ayahuasca), window 2 and 3 are considered as recordings under the influence of ayahuasca.

| Time window | Ingestion of ayahuasca at t = 0 minutes |
|:---:|:---:|
| 1 | -25 to 50 minutes |
| 2 | 50 to 125 minutes |
| 3 | 125 to 200 minutes |

defined as the "control". This is reasonable as it is known from [21], that the blood plasma concentration of the main psychedelic compound DMT is low until 50 minutes after ingestion. Windows two and three were both defined as thoroughly influenced by ayahuasca. The ayahuasca-influenced time series were divided into two windows to enhance the quantity of data points for the machine learning method. Even though the number of independent samples (subjects) did not change, increasing the data points by splitting the time series is a common machine learning approach [82, 83]. Even though the number of independent samples (subjects) did not change, increasing the data points by splitting the time series is a standard machine learning approach. Furthermore, in the following classification task, only two classes will be labeled class zero (without ayahuasca) and labeled class one (with ayahuasca). The scheme of this methodology is shown in Fig 1. All participants' EEG time series were successively combined and stored in a 2D matrix to feed the data into the machine learning algorithm. Each column represents an electrode, and each row represents the amplitude of each time point of the EEG signal. For each of the three time windows, a 2D matrix was constructed.

**2.3.2 Connectivity matrices.** The matrices of connectivity were calculated by the well known Pearson correlation. It is a widely used and successfully approved measure to capture the correlation of EEG electrodes [84–88].

The Pearson correlation was calculated for all electrode pairs resulting in three connectivity matrices per participant (for each time window). Fig 2 illustrates the workflow of this approach. The connectivity matrices were flattened into one vector to input the data into the machine learning algorithm. Then, all vectors were sequentially merged into a 2D matrix. Each column represents a connection between two brain regions, and each row represents a subject. Such a 2D matrix was generated for each of the three time windows.

**2.3.3 Complex network measures.** For each connectivity matrix (see subsection 2.3.2 Connectivity matrices), a graph was generated to extract different complex network measures. The complex network measures were stored in a matrix to input the data into the machine learning algorithm. Each column represents a complex network measure, and each row a subject. Such a 2D matrix was generated for each of the three time windows. The following complex network measures were calculated: Assortativity [89, 90], average path length (APL) [91], betweenness centrality (BC) [92], closeness centrality (CC) [93], eigenvector centrality (EC) [94], diameter [95], hub score [96], average degree of nearest neighbors [97] (Knn), mean degree [98], second moment degree (SMD) [99], entropy degree [100], transitivity [101, 102], complexity, k-core [103, 104], eccentricity [105], density [106], and efficiency [107]. Furthermore, newly developed metrics reflecting the number of communities in a complex network are used in this paper.

Furthermore, newly developed metrics reflecting the number of communities in a complex network are used in this paper. We perform the community detection algorithms to find the largest community, then calculate the average path length within this community and receive a single value as a result (that will be used to feed ML algorithm). The community detection algorithms used were:

- Fastgreedy community (FC) is defined in [108] as a hierarchical agglomerative clustering algorithm aimed at maximizing the modularity measure defined in [109].

- Infomap community (IC) Infomap community (IC) described in [110], the purpose behind this technique is to exploit the dynamics of random walks. This is accomplished by employing Huffman's method [111] and then calculating the minimization of the map equation to determine the number of communities [110].

- Leading eigenvector community (LC) is defined in [112]. It aims to calculate the eigenvector of the modularity matrix for the largest positive eigenvalue and then separate the vertices into two communities based on the sign of the corresponding element in the eigenvector.

- Label propagation community (LPC) is defined in [113]. It is an optimization algorithm [114] in which, at first, each node in the network has a label indicating its assignment, and then each node updates its label according to the label with the maximum number in its neighbors. This process is repeated until the network reaches a stable state and nodes with the same class are considered to belong to the same community. [115].

- Edge betweenness community (EBC) is defined in [109] is a divisive model based on the BC. At each iteration, this measure is calculated for all edges, and the one with the highest value of this measure is eliminated until the network contains N elements resulting in a hierarchical distribution of communities. The one with the highest modularity is adopted.

- Spinglass (SPC) is defined in [116] this algorithm considers the spin state of nodes as communities and tries to minimize the spin energy until it finds a ground state of the spin-glass model [117].

- Multilevel community (ML) Multilevel community (ML) is a greedy optimization method using modularity and is defined in [118].

Since the community detection algorithms were combined with the average path length, we extended the abbreviations by the letter "A" as follows: AFC, AIC, ALC, ALPC, AEBC, ASPC, and AMC.

Fig 3 depicts the entire workflow.

## 3 Results

The highest classification performance was obtained using the connectivity matrices with an accuracy of 92%, followed by the EEG time series (88%) and the complex network measures (83%) (see Table 2). The following subsections 3.1 EEG time series, 3.2 Connectivity matrices and 3.3 Complex network measures contain the results in more detail.

**Table 2. Performances of the SVM classifier for the different data types used in this paper.** The best performance is highlighted in bold. The classification of connectivity matrices best captured the changes in the brain due to ayahuasca.

| Type of data | Subset | AUC | Acc. | F1 score | Recall | Precision |
|---|---|---|---|---|---|---|
| **EEG time series** | **Train** | 0.87 | 0.89 | 0.88 | 0.87 | 0.89 |
| | **Test** | 0.85 | 0.88 | 0.86 | 0.85 | 0.86 |
| **Connectivity matrix** | **Train** | **0.92** | **0.94** | **0.93** | **0.92** | **0.96** |
| | **Test** | **0.88** | **0.92** | **0.90** | **0.88** | **0.94** |
| **Complex measure** | **Train** | 0.79 | 0.81 | 0.79 | 0.79 | 0.78 |
| | **Test** | 0.75 | 0.83 | 0.78 | 0.75 | 0.90 |

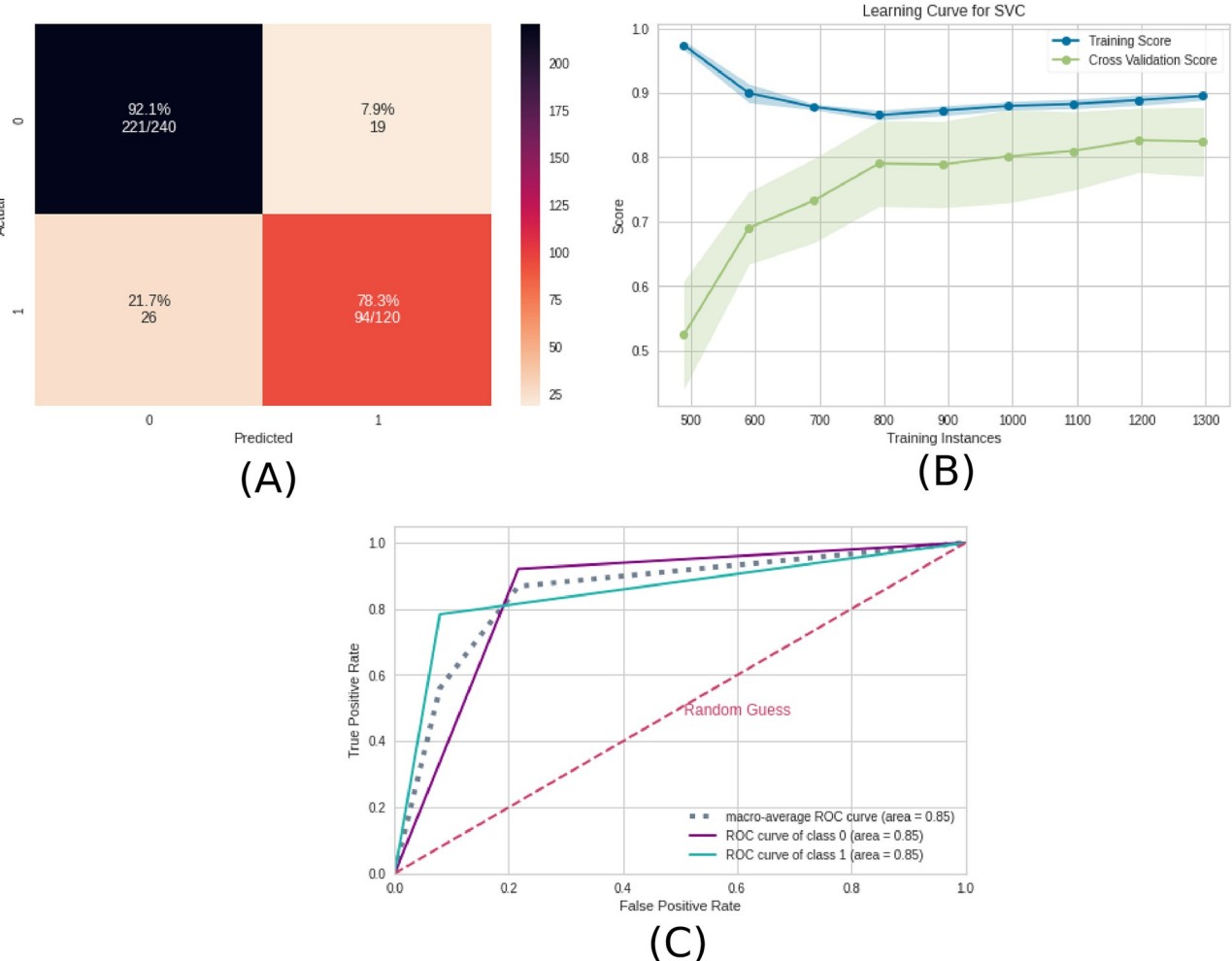

**Fig 4. Machine learning results using the EEG time series as input data.** A) Confusion matrix indicating a true negative rate of 92.1% (blue according to the color bar) and a true positive rate of 78.3% (orange according to the color bar). B) Learning curve for the training accuracy (blue) and for test accuracy (green). C) ROC curve of class 0 (without ayahuasca) and class 1 (with ayahuasca). The gray dotted curve is the macro-average accuracy (area under curve = 0.85) and the pink one the random classifier.

### 3.1 EEG time series

The performance of the test sample using the EEG time series was mean AUC of 0.85, mean precision of 0.88, mean F1 score of 0.86, mean recall of 0.85, and mean accuracy of 0.86. The precision measure is related to the positive class (with ayahuasca). Since the precision measure was slightly higher than the recall measure, the model can better detect the presence of ayahuasca instead of the absence of it.

In Fig 4, the confusion matrix (Fig 4A), the learning curve (Fig 4B), and the ROC curve (Fig 4C) are plotted.

The learning curve evaluates the predictability of the model by varying the size of the training set [45]. Fig 4B shows that the highest accuracy in the test sample can only be achieved when the entire database is used.

Not all electrodes of the EEG recording were equally important for the classification. According to the SHAP values, the most important region for the model was T7, located in the temporal region (see Fig 5). In order of importance, this region was followed by FC1, Fp1, P5,

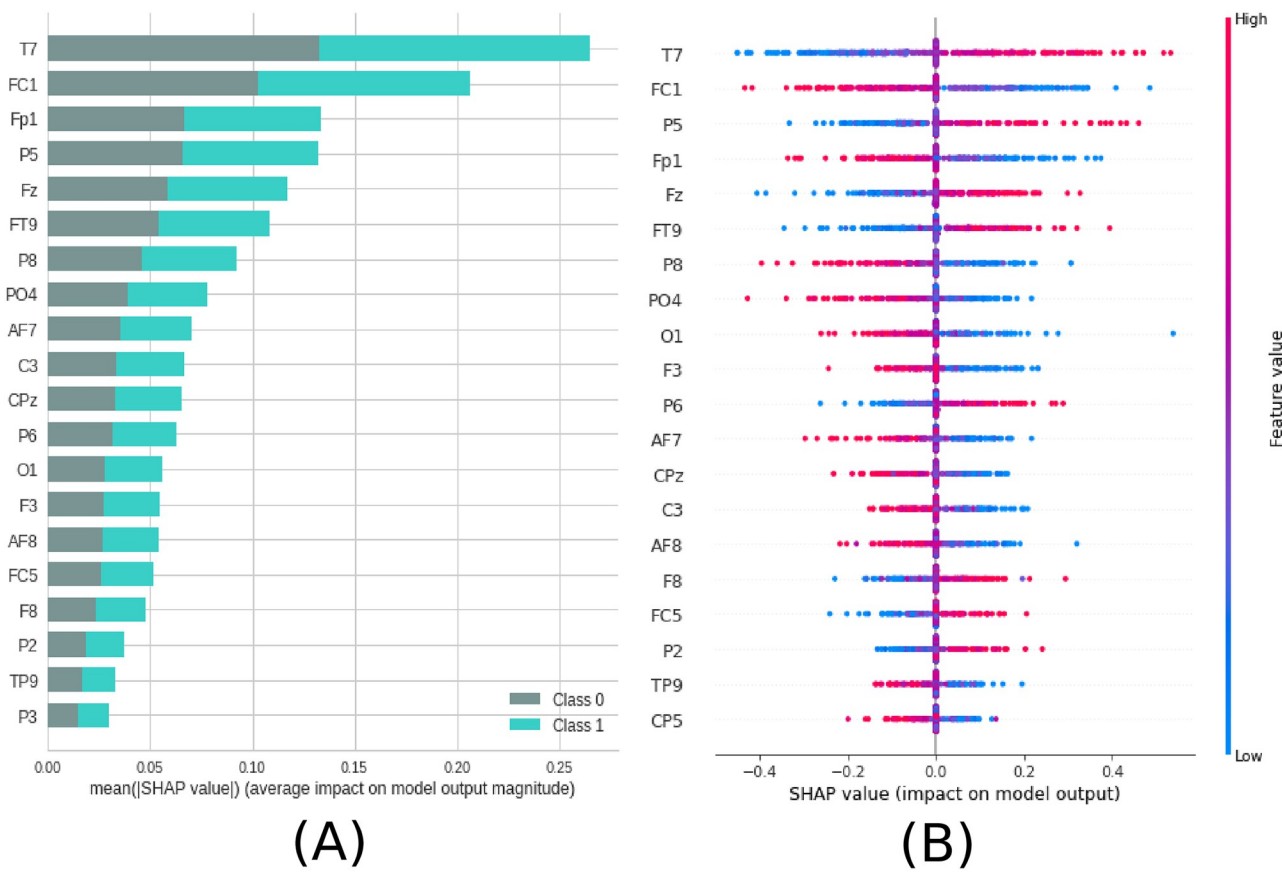

**Fig 5. Feature importance ranking for SVM classifier being the brain regions ranked in descending order of importance.** Brain region T7 is most important to classify the effect of ayahuasca. A) Feature ranking based on the average of absolute SHAP values over all subjects considering both classes (gray:without ayahuasca, cyan: with ayahuasca). B) Same as A) but additionally showing details of the impact of each feature on the model.

and Fz, located between frontal and central, frontal and parietal, parietal and frontal, respectively (see Fig 6A). In addition, Fig 6B shows details of the impact of each feature on the model. Positive SHAP values are shown when the presence of ayahuasca is detected, and negative SHAP values are shown when the absence of ayahuasca is detected. The colors indicate whether the feature value was low (blue) or high (red). Since the feature consists of the amplitudes of the EEG time series, it can be seen that for T7, the low amplitudes (blue dots) were important to detect the absence of ayahuasca (negative SHAP values), and the high amplitudes (red dots) were important to detect the presence of ayahuasca (positive SHAP values).

### 3.2 Connectivity matrices

For the connectivity matrices, the test sample performance was a mean AUC of 0.88, mean accuracy of 0.92, mean F1 score of 0.90, mean recall of 0.88, and mean precision of 0.94.

Similar to the previous subsection 3.1 EEG time series, the precision measure was higher than the recall measure and therefore the model can better detect the presence of ayahuasca. In Fig 7, the confusion matrix (Fig 7A), the learning curve (Fig 7B), and the ROC curve (Fig 7C) are plotted. Similar to EEG time series, the learning curve for the connectivity matrices shows that the highest accuracy in the test sample can only be achieved when the entire database is used.

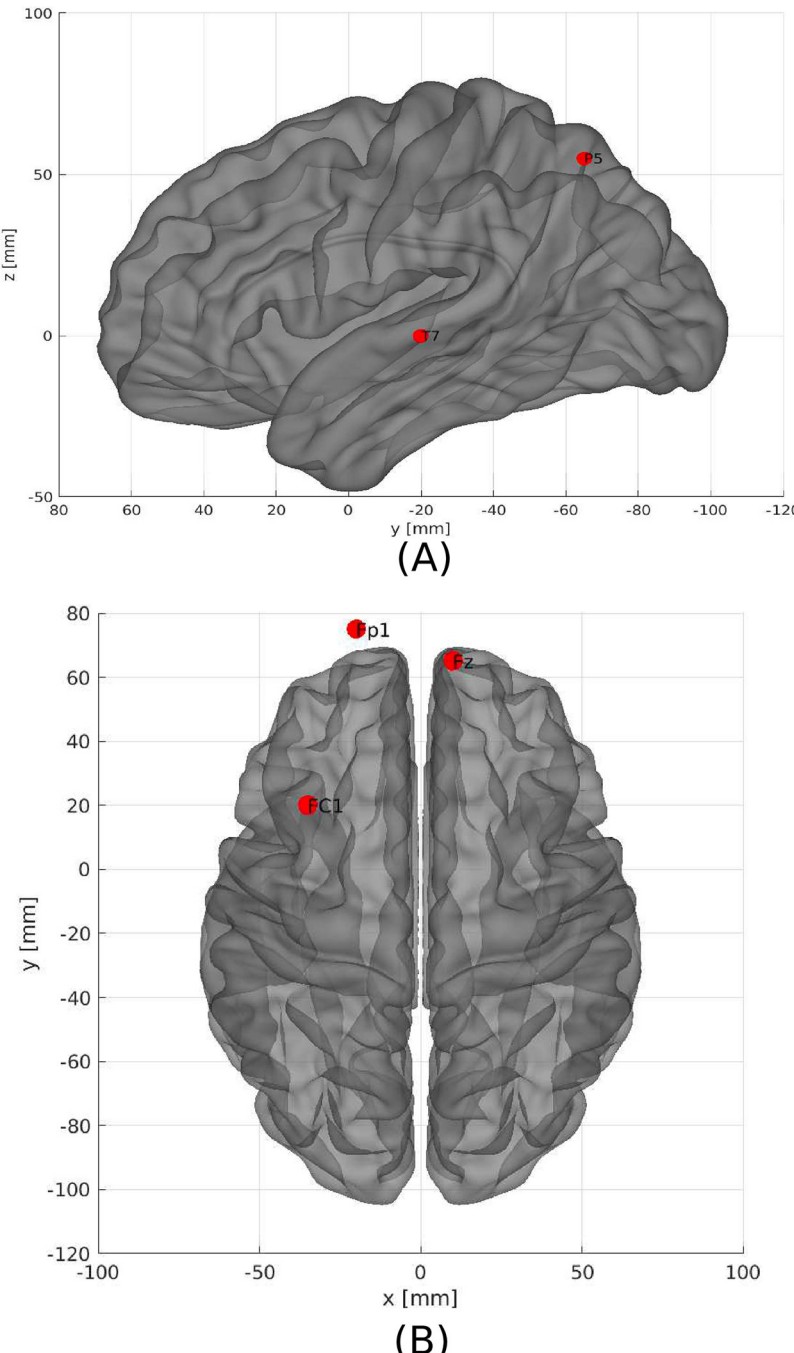

**Fig 6. The five most important brain regions considering EEG time series as input data.** A)—Sagittal left plane showing the brain region for the channel T7 and P5. B) Axial dorsal plane showing the brain regions Fz, Fp1 and FC1. The brain plot was made using Braph tool [119], based on the coordinates in [120, 121].

In order to reveal the importance of the brain connections, the SHAP values were used as in the preview subsection 3.1 EEG time series. The results are shown in Fig 8. From that the most important connection was between F3 (frontal region) and PO4 (between parietal and occipital region). In addition, in Fig 8B it can be seen that for the connection between F3 and

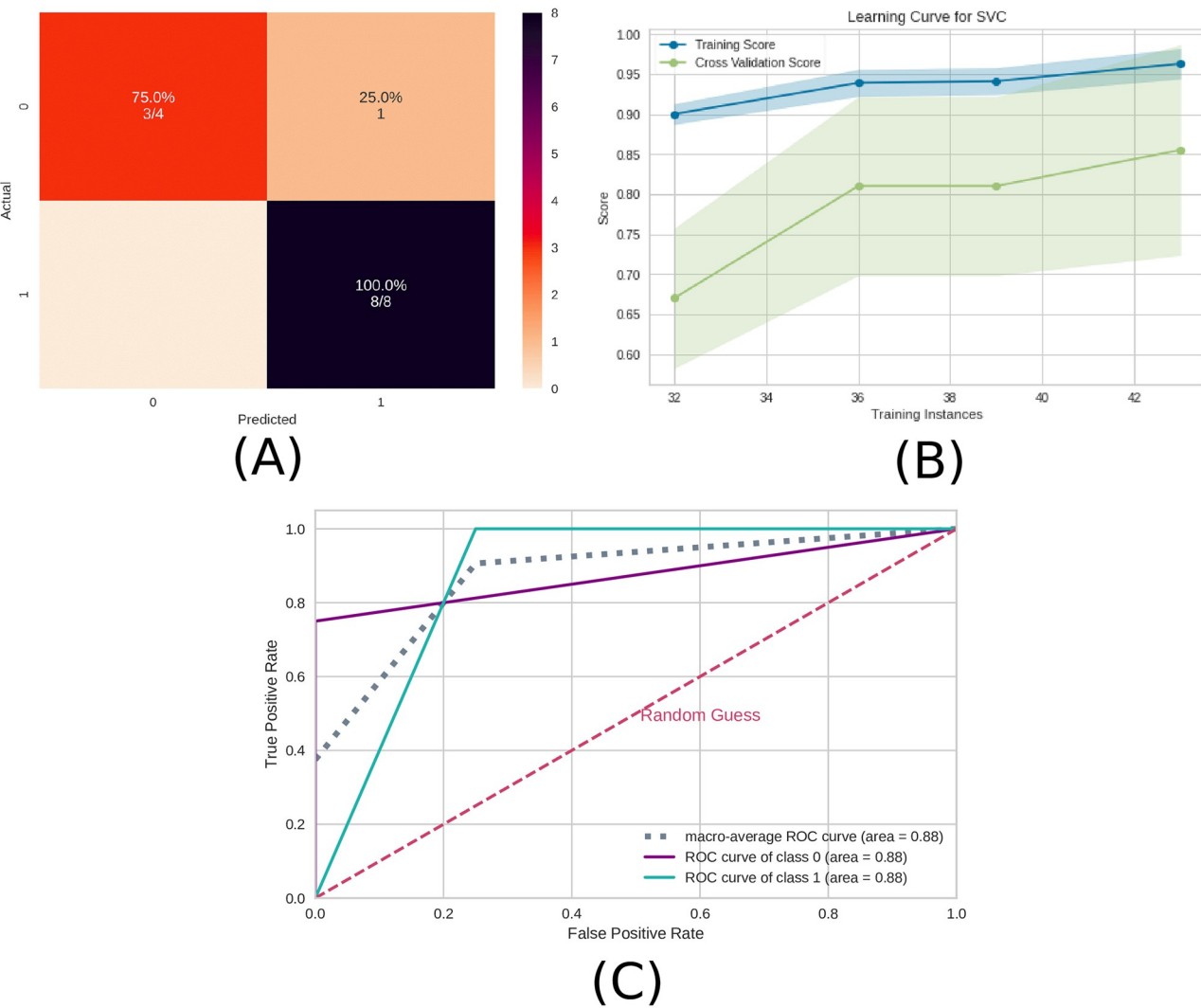

**Fig 7. Machine learning results using the connectivity matrices as input data.** A) Confusion matrix indicating a true negative rate of 75% (orange according to the color bar) and a true positive of 100% (blue according to the color bar). B) Learning curve for the training accuracy (blue) and for test accuracy (green). C) ROC curve of class 0 (without ayahuasca) and class 1 (with ayahuasca). The gray dotted curve is the macro-average accuracy (area under curve = 0.88) and the pink one the random classifier.

PO4, low values of correlation (blue dots) were important for detecting the absence of ayahuasca (negative SHAP values), and high values of correlation (red dots) were important for detecting the presence of ayahuasca (positive SHAP values).

The location in the brain can be seen in Fig 9.

## 3.3 Complex network measures

The test sample performance using the complex network measures was a mean AUC of 0.75, mean accuracy of 0.83, mean F1 score of 0.78, mean recall of 0.75, and mean precision of 0.90.

Similar to the previous subsections 3.1 EEG time series and 3.2 Connectivity matrices, the precision measure was higher than the recall measure, and therefore the model can better detect the presence of ayahuasca.

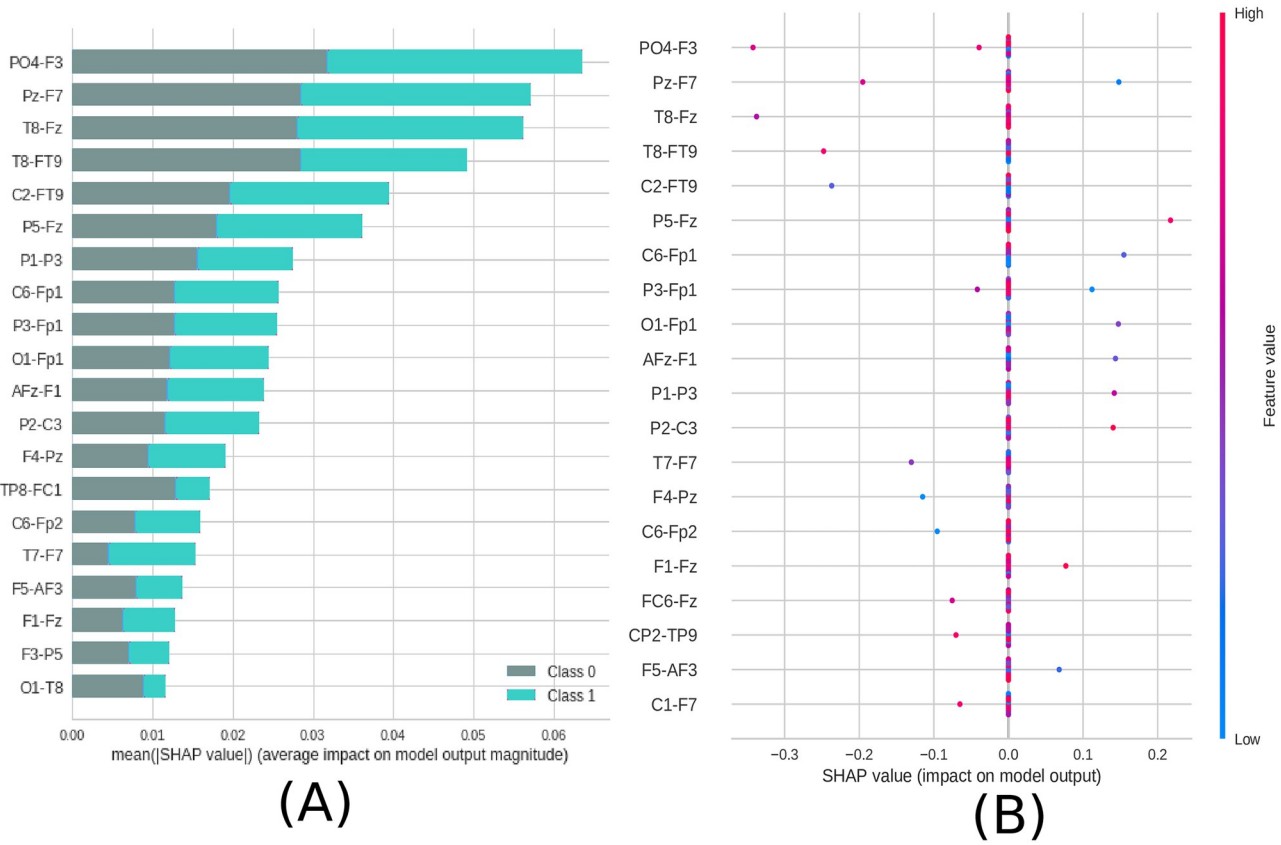

(A)          (B)

**Fig 8. Feature importance ranking for SVM classifier being the connections of brain regions ranked in descending order of importance.** The connection between the regions PO4 and F3 is the most important to classify the effect of ayahuasca. A) Feature ranking based on the average of absolute SHAP values over all subjects considering both classes (gray: without ayahuasca, cyan: with ayahuasca). B) Same as A) but additionally showing details of the impact of each feature on the model.

In Fig 10, the confusion matrix (Fig 10A), the learning curve (Fig 10B), and the ROC curve (Fig 10C) are plotted. Again, the entire database is necessary in order to get the highest accuracy.

From the SHAP values in Fig 11 it can be seen that the most important measure for the model was the CC, followed by assortativity, and the newly introduced measures ASC and ASPC. In addition, in Fig 11B can be seen that for the CC measure, low values of this metric (blue dots) were important for detecting the absence of ayahuasca (negative SHAP values), and high values of this metric (red dots) were important for detecting the presence of ayahuasca (positive SHAP values).

## 4 Discussion

In this paper, we aimed to answer the question if it is possible to automatically detect brain activity changes due to ayahuasca using machine learning and which features are most important and could act as biomarkers.

Our results show that it is possible to automatically detect the changes due to ayahuasca. The classification accuracy was above 75% for all three data abstraction levels. The classification accuracy of connectivity matrices was higher than the raw EEG time series, suggesting that connection changes are more important between brain regions than within brain regions.

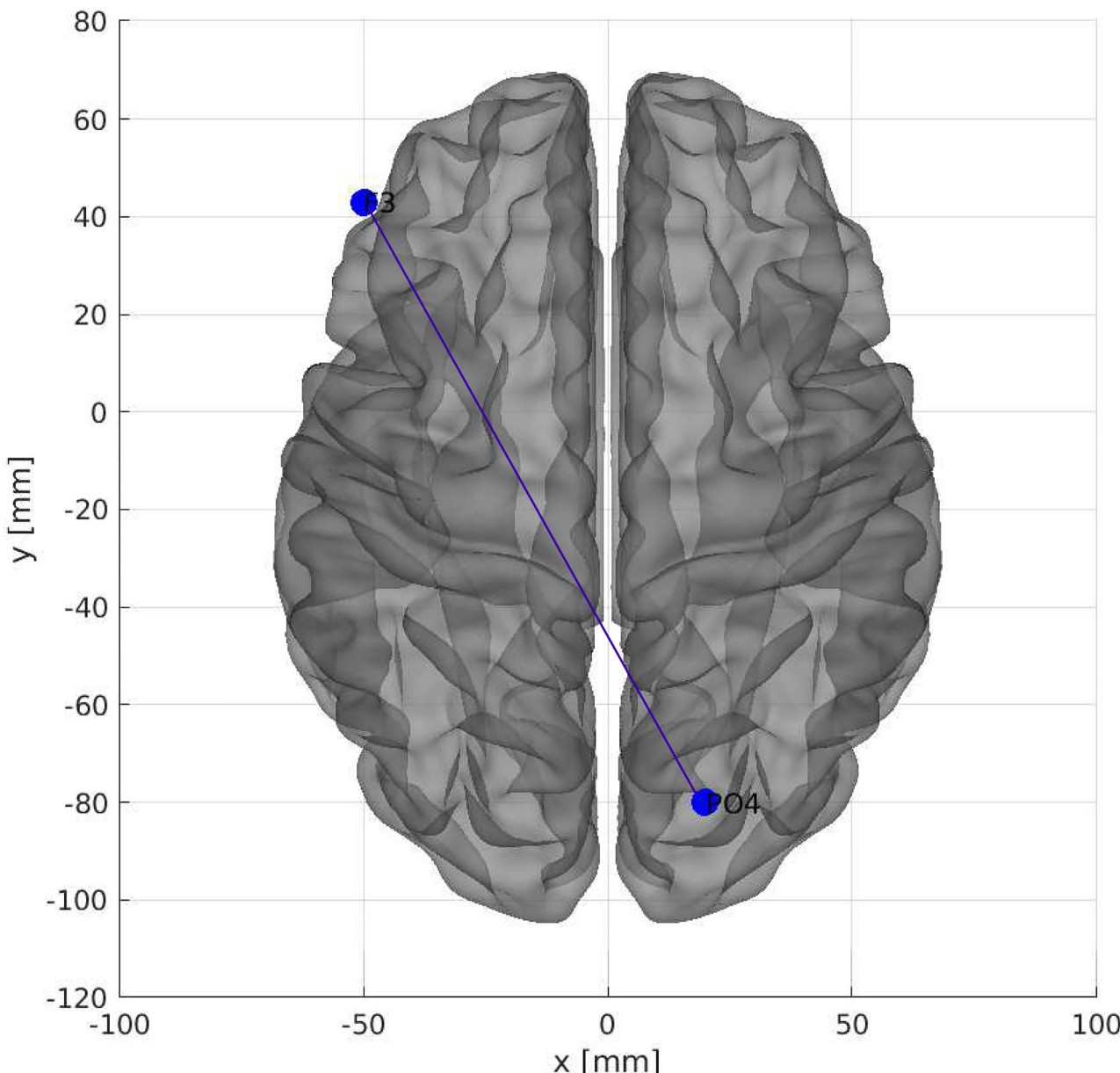

**Fig 9. The most important connection of brain regions considering connectivity matrices as input data.** Axial dorsal plane showing the brain regions connection between F3 and PO4. The brain plot was made using Braph tool [119], based on the coordinates in [120, 121].

This result is important since the connectivity matrices improved the accuracy and produced efficiency gains, such as reduced data storage and faster machine learning training. This would be especially useful for larger datasets, where raw time series may be very costly, for example, in hospital diagnosis systems.

## 4.1 EEG time series

The raw EEG time series analysis revealed that the frontal and the temporal lobe were the most affected brain regions. In line with that, studies using single photon emission computed

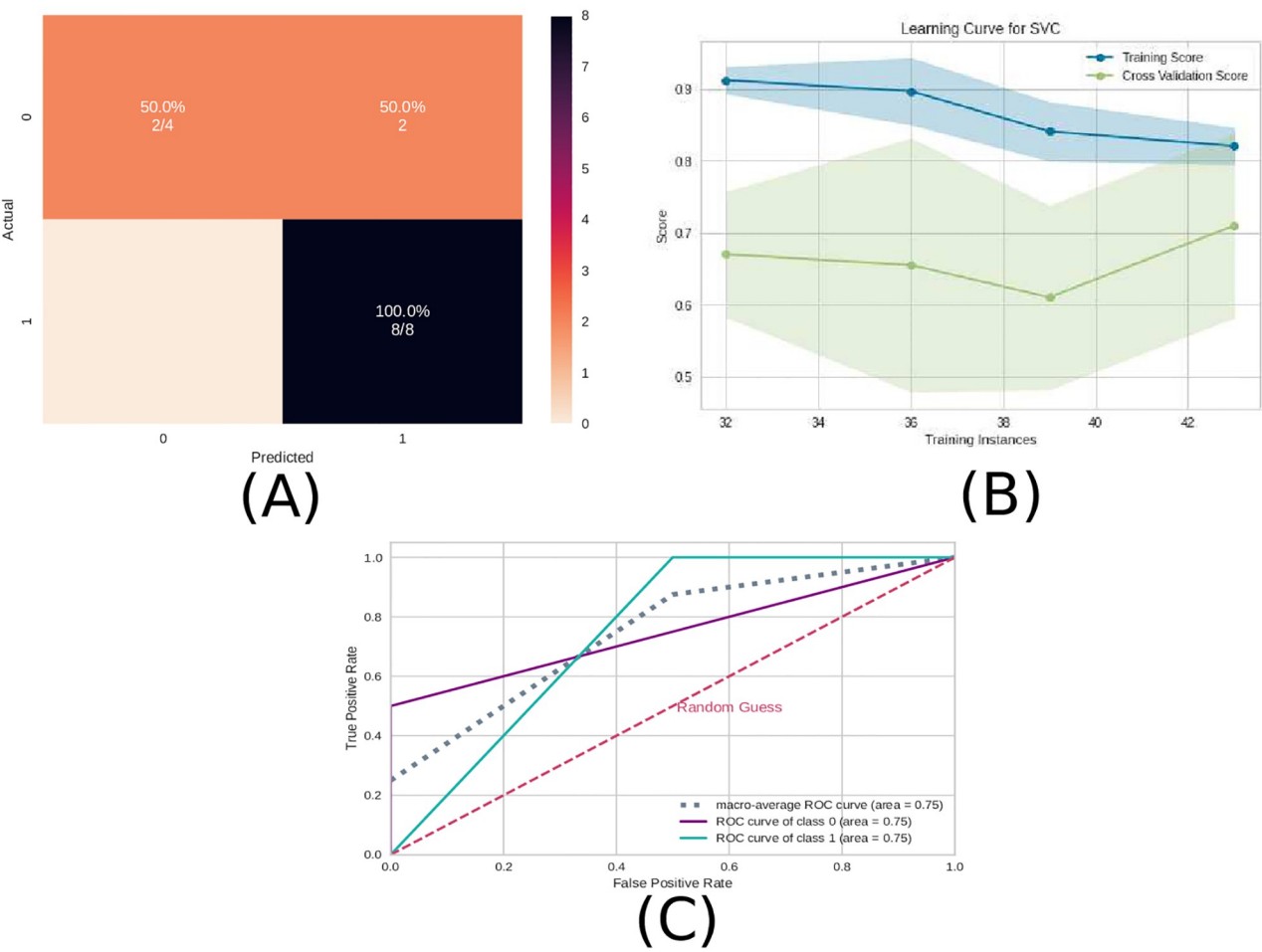

**Fig 10. Machine learning results using the complex network measures as input data.** A) Confusion matrix indicating a true negative rate of 50% (orange according to the color bar) and a true positive rate of 100% (blue according to the color bar). B) Learning curve for the training accuracy (blue) and for test accuracy (green). C) ROC curve of class 0 (without ayahuasca) and class 1 (with ayahuasca). The gray dotted curve is the macro-average accuracy (area under curve = 0.75) and the pink one the random classifier.

tomography (SPECT) have reported that ayahuasca increases blood perfusion in the frontal regions of the brain, more specifically, the insula, left nucleus accumbens, left amygdala, para-hippocampal gyrus, and left the subgenual area [16, 122]. Furthermore, works using functional magnetic resonance imaging have observed activation in the brain's occipital, temporal, and frontal areas [10, 123]. These regions are related to introspection, emotional processing, and the therapeutic effects of traditional antidepressants [124] and most interestingly, it may also affect motor and cognitive functions in other neurological disorders, such as Parkinson's disease and Alzheimer's disease, respectively [125, 126].

## 4.2 Connectivity matrices

The correlation between the left frontal cortex (F3) and right parietal-occipital (PO4) was most important in terms of brain connections.

[127] showed that synchronization in the gamma band between the parietal-occipital and frontal cortices was present during face recognition tasks. Since the EEG time series data used

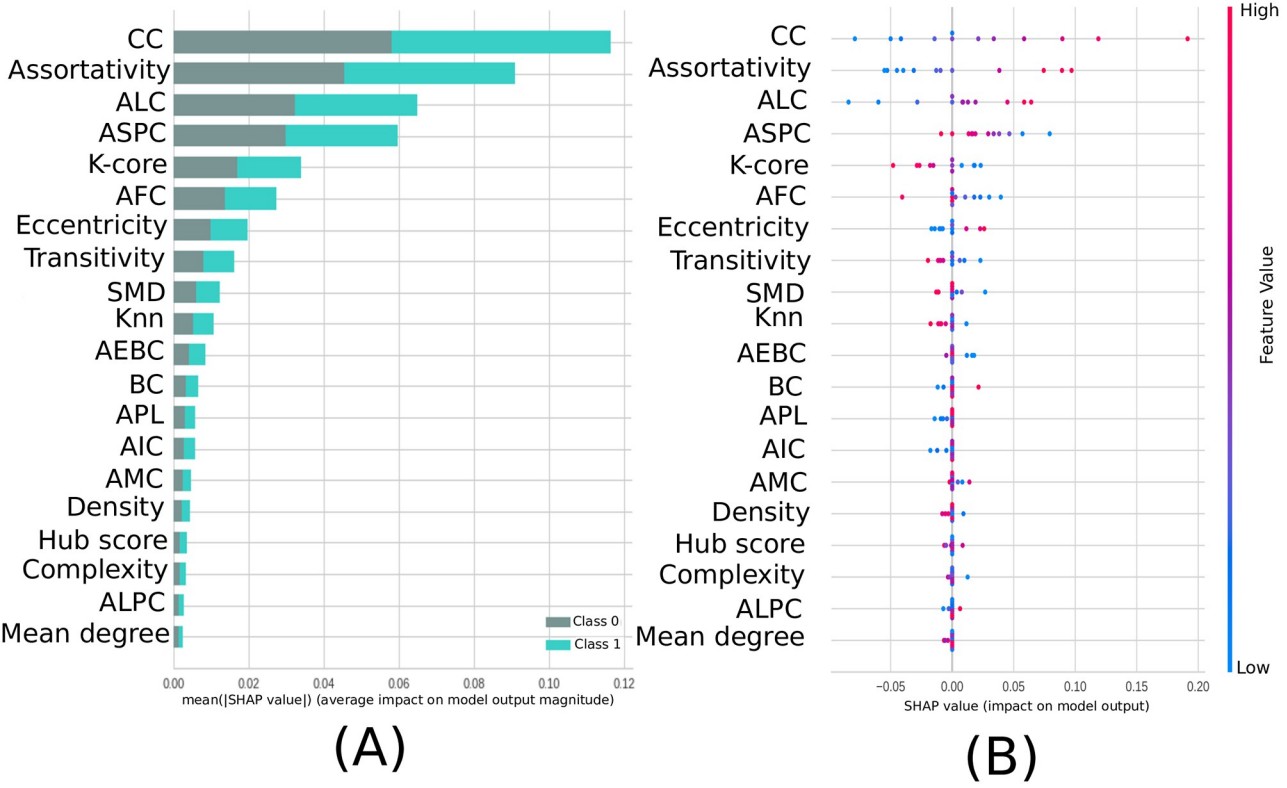

**Fig 11. Feature importance ranking for SVM classifier being the features ranked in descending order of importance.** The CC measure is the most important to classify the effect of ayahuasca. A) Feature ranking based on the average of absolute SHAP values over all subjects considering both classes (gray: without ayahuasca, cyan: with ayahuasca). B) Same as A) but additionally showing details of the impact of each feature on the model.

in this work only contained the gamma band, the P04-F3 connection could point to similar cognitive processes in the subjects during ayahuasca-mediated visual hallucinations.

### 4.3 Complex network measures

The most important complex network measure was CC. CC is a centrality measure that can be defined as the inverse of the average length of the shortest path from one node to all other nodes in the network [128]. The idea is that important nodes participate in many shortest paths within a network and, therefore, play an important role in the flow of information in the brain [93]. The CC was also the most important measure in other papers related to the differentiation of patients with AD [129–132]. In these papers, CC was shown to decrease due to AD disease, while ayahuasca ingestion increased the median value of this measure (see Fig 12).

The second most important complex network measure was assortativity. This measure refers to the resilience of networks [90]. A positive assortativity coefficient indicates a network with a resilient core due to the interconnected nodes of high degree [128]. This measure was also associated with AD in several works [133, 134] whose results showed an increase in the assortativity value in contrast to what was found here, where with the use of ayahuasca, the assortativity value (median) decreased (see on Fig 12). It should be noted that although the median value decreased, the upper confidence interval of the distribution increased.

In summary, the results suggest a possible relationship between ayahuasca and AD in terms of the brain network, indicating a therapeutic potential. Indeed, a possible mechanism of how

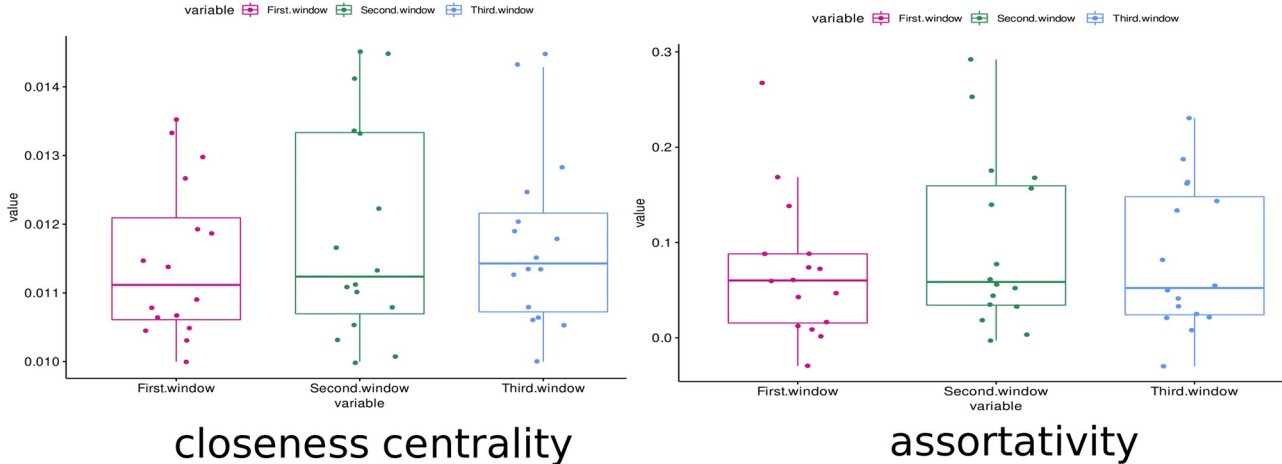

**Fig 12. Boxplot of the closeness centrality and assortativity measures.** These measures were calculated for all subjects in the first, second and third windows (respectively in pink, green and blue). It can be seen that the median of the closeness centrality measure (central bar in the boxplot) increased with the use of ayahuasca. The median of the assortativity, in contrast, decreased with the use of ayahuasca.

ayahuasca acts against AD was described in [19]. According to this, the ayahuasca compound dimethyltryptamine (DMT) agonizes the sigma 1 receptor (Sig-1R) and thereby regulates endoplasmic reticulum (ER) stress and Unfolded Protein Response (UPR), which are thought to play a crucial role in neuropsychiatric diseases such as AD.

The seven measures developed here concerning community detection are ranked among the twenty most important measures for classification, with ALC ranking third (see Fig 11). ALC is associated with the size of the largest community found by the leading eigenvector community (LC) detection algorithm. This metric shows increased values (compared to controls) in communities with larger path lengths after the use of ayahuasca (Fig 11B), indicating communities with larger paths after using this psychedelic. Larger brain communities were also found in [135] after the use of ayahuasca. There are two contrasting concepts in the brains of large vertebrates: functional segregation (or specialization) and integration (or distributed processes) [136, 137]. Larger communities also indicate that the balance between functional segregation and integration in the brain was disrupted. This suggests that the distribution of information is slower.

Overall, the classification was successful by considering the complete set of measures rather than just one single measure. As shown in Fig 12, even the most important measures CC and assortativity, did not show much difference between the first window (without ayahuasca) and the other windows (with ayahuasca). Together with the other less important measures, however, the machine learning method was able to distinguish both classes successfully. This leads to the conclusion that a single feature is insufficient as a biomarker, while the different features used in this work may serve as a biomarker.

## 5 Conclusion

In summary, the results obtained in our study demonstrated that the application of machine learning methods was able to detect changes in brain connectivity during ayahuasca use automatically. Additionally, we demonstrated that the connectivity matrices are the best abstraction level to detect brain changes caused by this psychedelic.

At level abstraction A, our findings suggest that this substance affects important brain regions related to cognitive, psychiatric, and motor functions. These effects may alleviate different symptoms of diseases affecting the brain.

At level abstraction B, the connection between F3 and PO4 is the most important while using ayahuasca according to our classifier model, a significant discovery in psychedelic literature. This connection may point to a cognitive process similar to face recognition in individuals during ayahuasca-mediated visual hallucinations.

Concerning the complex network measures at level abstraction C, CC, assortativity, and one of the new measures developed here, ALC, capture the best brain changes caused by ayahuasca. The new ALC measure inferred that larger communities are associated with this psychedelic and the opposite in its absence. Larger communities suggest that the distribution of information is slower with the use of this substance. Therefore, the present study's findings support that cortical brain activity becomes more entropic under psychoactive substances. [138–140]. There is, however, evidence that psychedelics do not simply make the brain more random, but after the typical organization of the brain is disrupted, strong and topologically far-reaching functional connections emerge which are not present in the natural state of mind.

While our methodology has proven effective, it is focused on the acute evaluation of psychedelics. Consequently, more research is necessary to determine how psychedelics affect the functional connectivity of the brain over the long term using our workflow.

In summary, we have developed a robust computational workflow that provides insights into the mechanism of action of ayahuasca and the interpretability of how it modifies brain networks.

Finally, the same methodology applied here may help interpret EEG time series from patients who consumed other psychedelic drugs, such as pure DMT [141]. In future work, we aim to apply this workflow to recordings from our laboratory using in vitro neuronal networks on microelectrode arrays to study the effects of psychedelics at a single network level. Thus, regardless of the equipment used to collect the data, we would like to verify whether the same method used here can detect changes due to different psychedelics.

## Supporting information

**S1 Appendix.**
(PDF)

## Author Contributions

**Conceptualization:** Caroline L. Alves, Francisco A. Rodrigues, Manuel Ciba.

**Data curation:** Caroline L. Alves, Francisco A. Rodrigues, Manuel Ciba.

**Formal analysis:** Caroline L. Alves, Rubens Gisbert Cury, Kirstin Roster, Aruane M. Pineda, Francisco A. Rodrigues, Manuel Ciba.

**Funding acquisition:** Francisco A. Rodrigues, Christiane Thielemann.

**Investigation:** Caroline L. Alves, Rubens Gisbert Cury, Kirstin Roster, Aruane M. Pineda, Francisco A. Rodrigues, Christiane Thielemann, Manuel Ciba.

**Methodology:** Caroline L. Alves.

**Project administration:** Christiane Thielemann, Manuel Ciba.

**Resources:** Francisco A. Rodrigues, Christiane Thielemann.

**Supervision:** Francisco A. Rodrigues, Christiane Thielemann, Manuel Ciba.

**Validation:** Caroline L. Alves, Rubens Gisbert Cury, Kirstin Roster, Aruane M. Pineda, Francisco A. Rodrigues, Manuel Ciba.

**Visualization:** Caroline L. Alves.

**Writing – original draft:** Caroline L. Alves, Rubens Gisbert Cury, Kirstin Roster, Francisco A. Rodrigues, Christiane Thielemann, Manuel Ciba.

**Writing – review & editing:** Caroline L. Alves, Aruane M. Pineda.

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
