## [Decision Letter · Decision Letter 0]

26 Aug 2022

PONE-D-22-21178Application of machine learning and complex networkmeasures to an EEG dataset from ayahuasca experimentsPLOS ONE

Dear Dr. Alves,

Thank you for submitting your manuscript to PLOS ONE. After careful consideration, we feel that it has merit but does not fully meet PLOS ONE’s publication criteria as it currently stands. Therefore, we invite you to submit a revised version of the manuscript that addresses the points raised during the review process.

We look forward to receiving your revised manuscript.

Kind regards,

Yiming Tang, Ph.D.

Academic Editor

PLOS ONE

Journal Requirements:

2. Please ensure that you have specified (1) whether consent was informed and (2) what type you obtained (for instance, written or verbal, and if verbal, how it was documented and witnessed). If your study included minors, state whether you obtained consent from parents or guardians. If the need for consent was waived by the ethics committee, please include this information.

"No"

"NO authors have competing interests"

Reviewers' comments:

Reviewer's Responses to Questions

**Comments to the Author**

1. Is the manuscript technically sound, and do the data support the conclusions?

Reviewer #1: Partly

Reviewer #2: Yes

Reviewer #3: Partly

Reviewer #4: Yes

2. Has the statistical analysis been performed appropriately and rigorously? 

Reviewer #1: Yes

Reviewer #2: N/A

Reviewer #3: Yes

Reviewer #4: Yes

3. Have the authors made all data underlying the findings in their manuscript fully available?

Reviewer #1: Yes

Reviewer #2: No

Reviewer #3: Yes

Reviewer #4: Yes

4. Is the manuscript presented in an intelligible fashion and written in standard English?

Reviewer #1: Yes

Reviewer #2: Yes

Reviewer #3: Yes

Reviewer #4: Yes

5. Review Comments to the Author

Reviewer #1: The Authors analyse a publicly available EEG dataset with various machine learning techniques in order to describe the effect of a drug called "ayahuasca" on functional connectivity networks of the brain. Overall, the findings are interesting: they indicate the potential therapeutic use of ayahuasca.

My major concern, because of which I answered the question about technical soundness with "partially" is that in order to assist reproduction and independent validation of the results, and to promote the use of the same methodology in other studies, I think the Authors should publish their implementation of the methods, including preprocessing steps (i.e., source codes of the software they implemented during their research work).

As for the language, the Authors should keep in mind that PLOS ONE does not copyedit manuscripts, therefore they should proofread their work carefully. For example, special attention should be payed to references, both references to subsections (that are often broken in the current version of the manuscript) and references to the literature. See for example: "...known from schenberg2015acute" (third line in Section 3.1.1).

As for the methodology, while I understand that the Authors used standard machine learning methods, both for classification of the data, as well as for the other subtasks, such as community detection, it would be worth to discuss recent approaches that were designed to deal with brain activity data. See, for example:

http://biointelligence.hu/pdf/saci2020_5_buza.pdf

https://link.springer.com/chapter/10.1007/978-3-319-45823-6_59

Reviewer #2: The paper can be considered for publication if it is improved,

- the current version is not at the journal level.

- difficult to understand.

-In the related work section, the authors have added short comments to new references; among these, more details of the method proposed in the above-cited paper have to be reported. Should be better explained to outline the limitations of the approaches.

-In the Literature review section, the authors should find out the difference between the proposed model and the existing methods should be further described.

-The paper's presentation should be improved. It is difficult to understand.

-Your abstract does not highlight the specifics of your research or findings.

-The introduction should be rewritten to clarify its message. Drawbacks of former proposals should be clearly indicated and innovations and new ideas highlighted.

-Problem statement and objective are not clear. The authors should give a more accurate description of the existing methods.

-Tables & figures present a lot of statistics but need a more detailed explanation (which is missing in the text).

-I feel that more explanation would be needed on how the proposed method is performed

-If no one has proposed a method like the proposed algorithm, this claim should be highlighted much more. Else, it should be indicated who has done this, and it should be indicated what the innovations of the current paper are.

- Improve text formatting.

-why have not been used other DL based methods

-Please, also provide a paragraph with three to five clear positive impacts of your method.

- the authors should further detail the preparation of the dataset.

-the figures must be in better format & resolution

-How to set the parameters of DL algorithms.

-There are no real insightful conclusions drawn from the study and no suggestions for practical use of the results. Therefore, the conclusion section should be totally rewritten in order to:

- You must more clearly highlight the theoretical and practical implications of your research

-Discuss research contributions, Indicate practical advantages, and discuss research limitations.

-supply solid and insightful future research suggestions, and references must be updated.

Reviewer #3: The authors studied brain activity changes under ayahuasca by using machine learning and complex network features and showed the high performance of SVM to predict the brain activity change. The research topic is interesting, and the manuscript is written well. However, I have major concerns about the method and results.

Major comments

- Is this the first study to investigate brain activity under ayahuasca? If not so, the authors should introduce them in Introduction section. Even if so, other previous studies on relationship between brain characteristics (e.g., brain activity, functional connectivity) and medical herbs should be cited in Introduction. Moreover, the authors should describe what kind of results would be expected based on the previous studies.

- Why did some ML methods fail to predict test data? For example, XGBoost had remarkably low performance for test dataset.

- The authors did not analyze control experiment data. I’m wondering if the result can be obtained even in without-drug conditions. For example, the authors cannot exclude a possibility that the prediction was based on the characteristics along time (e.g., change due to tiredness).

Minor comments

- 3.2.1: LG -> LR

- Abstract: remove new lines

Reviewer #4: Thank you for providing opportunity to review this manuscript. Outcome of the study is interesting. The general impression of this manuscript should follow proper order. Such as:

1. In introduction(page2), it would be nice the research questions are in sentences not in bullet form.

2. Data (page2) should move to the Material and Methods (in page3).

3. The biggest concern is that methodology of the data collection utilized. 16 healthy makes and females patients were selected for the study. How were these healthy patients selected? 32 patients were appropriate number? what criteria was used to decide healthy patients? What method of sampling used to select patients?

4. What about the ethical consideration of this study? Though it is mentioned in the Ethics Statement, it would be nice to declare in the manuscript.

5. Figures are shown after the references, which is troublesome for the reader. I would advise authors to keep some important figures body of the writing and remaining should go to the bottom of the manuscript.

6. PLOS authors have the option to publish the peer review history of their article (what does this mean?). If published, this will include your full peer review and any attached files.

Reviewer #1: No

Reviewer #2: No

Reviewer #3: No

Reviewer #4: No

---

## [Author Response · Author response to Decision Letter 0]

7 Oct 2022

Dear all,

We would like to thank the reviewer for their careful consideration of our manuscript and recommendation

for acceptance after major revision. The comments have been very useful and have helped us improve the analysis and clarity of explanation in the manuscript. We provide point-by-point responses to each comment

in the 'Response to Reviewers' file.

Best regards,

Caroline L. Alves

---

## [Decision Letter · Decision Letter 1]

24 Oct 2022

Application of machine learning and complex network

measures to an EEG dataset from ayahuasca experiments

PONE-D-22-21178R1

Dear Dr. Alves,

We’re pleased to inform you that your manuscript has been judged scientifically suitable for publication and will be formally accepted for publication once it meets all outstanding technical requirements.

Kind regards,

Yiming Tang, Ph.D.

Academic Editor

PLOS ONE

Additional Editor Comments (optional):

Reviewers' comments:

Reviewer's Responses to Questions

**Comments to the Author**

1. If the authors have adequately addressed your comments raised in a previous round of review and you feel that this manuscript is now acceptable for publication, you may indicate that here to bypass the “Comments to the Author” section, enter your conflict of interest statement in the “Confidential to Editor” section, and submit your "Accept" recommendation.

Reviewer #1: All comments have been addressed

Reviewer #2: All comments have been addressed

Reviewer #3: All comments have been addressed

Reviewer #4: All comments have been addressed

2. Is the manuscript technically sound, and do the data support the conclusions?

Reviewer #1: Yes

Reviewer #2: Yes

Reviewer #3: Yes

Reviewer #4: Yes

3. Has the statistical analysis been performed appropriately and rigorously? 

Reviewer #1: Yes

Reviewer #2: Yes

Reviewer #3: Yes

Reviewer #4: Yes

4. Have the authors made all data underlying the findings in their manuscript fully available?

Reviewer #1: Yes

Reviewer #2: No

Reviewer #3: Yes

Reviewer #4: Yes

5. Is the manuscript presented in an intelligible fashion and written in standard English?

Reviewer #1: Yes

Reviewer #2: Yes

Reviewer #3: Yes

Reviewer #4: Yes

6. Review Comments to the Author

Reviewer #1: The Authors addressed my concernes adequately.

Please double check the manuscript for typos (e.g. "avaiable" instead of "available" in footnote 1).

Reviewer #2: thanks for improving the paper based on the comments.

the current version can be considered for publication.

Reviewer #3: (No Response)

Reviewer #4: Most of the comments addressed by the author. This manuscript is ready for the publication. I have a request that it would be nice if title Data of introduction moved to 3. material and methods.

7. PLOS authors have the option to publish the peer review history of their article (what does this mean?). If published, this will include your full peer review and any attached files.

Reviewer #1: No

Reviewer #2: No

Reviewer #3: No

Reviewer #4: **Yes: **Shreedhar Acharya
